# Extracorporeal Shock Wave Therapy for the Treatment of Musculoskeletal Pain: A Narrative Review

**DOI:** 10.3390/healthcare11212830

**Published:** 2023-10-26

**Authors:** Hortensia De la Corte-Rodríguez, Juan M. Román-Belmonte, Beatriz A. Rodríguez-Damiani, Aránzazu Vázquez-Sasot, Emérito Carlos Rodríguez-Merchán

**Affiliations:** 1Department of Physical Medicine and Rehabilitation, La Paz University Hospital, Paseo de la Castellana 261, 28046 Madrid, Spain; 2IdiPAZ Institute for Health Research, 28046 Madrid, Spain; 3Department of Physical Medicine and Rehabilitation, Cruz Roja San José y Santa Adela University Hospital, 28003 Madrid, Spain; juanmaromanbelmonte@gmail.com (J.M.R.-B.);; 4Medical School, Alfonso X El Sabio University, 28691 Madrid, Spain; 5Department of Orthopedic Surgery, La Paz University Hospital, 28046 Madrid, Spain; 6Osteoarticular Surgery Research, Hospital La Paz Institute for Health Research—IdiPAZ (La Paz University Hospital—Medical School, Autonomous University of Madrid), 28046 Madrid, Spain

**Keywords:** extracorporeal shock wave therapy, musculoskeletal conditions, musculoskeletal pain

## Abstract

Extracorporeal shock waves are high-intensity mechanical waves (500–1000 bar) of a microsecond duration with a morphology characterized by a rapid positive phase followed by a negative phase. Background: Extracorporeal shock waves have been used for pain treatment for various sub-acute and chronic musculoskeletal (MSK) problems since 2000. The aim of this article is to update information on the role of extracorporeal shock wave therapy (ESWT) in the treatment of various pathologies that cause MSK pain. Methods: Given that in the last two years, articles of interest (including systematic reviews and meta-analyses) have been published on less known indications, such as low back pain, nerve entrapments, osteoarthritis and bone vascular diseases, a literature search was conducted in PubMed, the Cochrane Database, EMBASE, CINAHL and PEDro, with the aim of developing a narrative review of the current literature on this topic. The purposes of the review were to review possible new mechanisms of action, update the level of evidence for known indications and assess possible new indications that have emerged in recent years. Results: Although extracorporeal shock waves have mechanical effects, their main mechanism of action is biological, through a phenomenon called mechanotransduction. There is solid evidence that supports their use to improve pain in many MSK pathologies, such as different tendinopathies (epicondylar, trochanteric, patellar, Achilles or calcific shoulder), plantar fasciitis, axial pain (myofascial, lumbar or coccygodynia), osteoarthritis and bone lesions (delayed union, osteonecrosis of the femoral head, Kienbock’s disease, bone marrow edema syndrome of the hip, pubis osteitis or carpal tunnel syndrome). Of the clinical indications mentioned in this review, five have a level of evidence of 1+, eight have a level of evidence of 1−, one indication has a level of evidence of 2− and two indications have a level of evidence of 3. Conclusions: The current literature shows that ESWT is a safe treatment, with hardly any adverse effects reported. Furthermore, it can be used alone or in conjunction with other physical therapies such as eccentric strengthening exercises or static stretching, which can enhance its therapeutic effect.

## 1. Introduction

Shock wave therapy is a non-invasive therapeutic procedure in which a single-impulse transient acoustic wave of 1 microsecond duration is applied to different target body regions to produce analgesia and facilitate healing through a mechanism called mechanotransduction [1]. Shock waves are considered an effective, non-invasive and cost- and time-efficient treatment [2].

Extracorporeal shock wave therapy (ESWT) was first used in 1980, and its indications were extended from lithotripsy to the treatment of delayed bone healing to chronic pain management in 2000 [3]. Plantar fasciitis was the first orthopedic disease to obtain FDA (Food and Drug Administration) approval for shock wave management [4], with many other indications subsequently being added. 

Although the physiological mechanisms of its therapeutic effect are not fully understood, shock wave therapy produces a biological effect (mechanotransduction) in its passage through the tissues, achieving an analgesic, osteogenic, neovascular and tissue repair effect. Among the beneficial effects they produce, it is worth highlighting that they produce analgesia, facilitate protein synthesis, increase vascularization, improve cell proliferation, produce calcium destruction in tissue and have a protective effect on cartilage and bone [5]. This has led to their use in numerous pathologies that cause chronic pain such as chronic tendinopathies, myofascial pain, osteoarthritis, nonunion fracture, bone vascular diseases and entrapment neuropathies. In the last two years, 12 articles of interest (including systematic reviews and meta-analyses) have been published on less known indications such as low back pain, nerve entrapment, osteoarthritis or bone vascular diseases. This has made it advisable to update the recommendations and effectiveness of shock waves in these indications.

The aim of this article is to update information on the role of ESWT in the treatment of various pathologies that cause musculoskeletal (MSK) pain, such as tendinopathies, low back pain, osteoarthritis, osteonecrosis, bone vascular diseases and nerve entrapments.

## 2. Materials and Methods

A literature search was conducted on 11 August 2023 in PubMed, the Cochrane Database, EMBASE, CINAHL and PEDro, using the keywords “extracorporeal shock wave therapy musculoskeletal pain”, with the aim of developing a narrative review of the current literature on this topic. Due to the large number of references published on the topic (3517), a selection was made in the literature to provide a synthesis based on the strength of the evidence. In those pathologies where there were papers with different levels of evidence, those with high levels of evidence (meta-analyses, systematic reviews and high-quality randomized clinical trials) were selected over those with lower levels of evidence (non-randomized trials, cohort studies, case–control studies, case series or expert opinions). In those pathologies for which no high-evidence studies had been published, low-evidence studies were included to provide a synthesis of the available clinical information. A total of 1745 articles were found in PubMed, 1461 in EMBASE, 195 in CINAHL, 115 in PEDro and 1 in the Cochrane Database, making a total of 3517 articles identified. After identifying duplicate articles, 149 records were screened. 

A total of 115 articles were considered of interest because they were closely related to the topic of this article, and 98 articles were finally included in this review. The following articles were included: (1) articles with a high level of evidence, including meta-analyses, systematic reviews and randomized clinical trials; (2) articles published within the last five years and able to provide recent information that was not included in other papers in the literature; (3) articles that deal with less frequent indications of shock waves such as nerve entrapment or bone vascular diseases; (4) articles that detail the protocol of the application of shock waves and that could be used to offer homogeneous clinical recommendations. The remaining articles (34) were excluded (Figure 1). Articles using shock waves in non-musculoskeletal pathologies, such as erectile dysfunction or lymphoedema, were excluded. Articles with a low level of evidence were also excluded when there were references with a high level of evidence (meta-analyses, systematic reviews and randomized clinical trials) on the same clinical indication.

Of the authors who contributed to this paper, HDLC-R and JMR-B conducted the initial literature search and wrote the first draft of the article. ECR-M, BAR-D and AV-S reviewed the draft and contributed to the final manuscript.

## 3. Results and Discussion

The results and critical discussion of the studies included in the narrative review are presented below. The following paragraphs refer to the mechanism of action of shock waves, their method of application and their effect on pain in different MSK processes. 

### 3.1. Mechanism of Action

Shock waves are rapidly generated pressure waves that travel faster than the speed of sound in the same medium through which the wave propagates. The mechanism of action by which shock waves produce their therapeutic effects is not yet fully understood [6].

The shock wave has two phases, an initial phase of rapid rise (<10 ns) with focus pressures of 50 to 80 MPa, followed by a relatively slow phase (milliseconds) of negative pressure of up to 10 MPa [7]. The curve generated by a shock wave is reflected in Figure 2. The shock wave produces a phenomenon called acoustic cavitation that could be largely responsible for its biological effects [6].

There are two types of shock waves. Focal shock waves, produced by electrohydraulic, piezoelectric or electromagnetic generators, consist of a negative phase and achieve much higher intensity on a target deep in the tissue to be treated. So-called radial waves are lower-intensity pressure waves generated via a pneumatic mechanism, without a negative phase, and whose highest intensity point is produced on the surface of the applicator [8]. When applied, the waves dissipate energy as they pass through tissues with different acoustic impedances, causing a release of kinetic energy that can activate tissue-reparative processes. The therapeutic effects include analgesic, osteogenic and tissue-reparative effects mediated by different mechanisms, as shown in Table 1 [9,10,11,12,13,14,15,16,17,18,19,20,21,22,23].

In summary, shock waves produce a decrease in pain, facilitate tissue repair and promote bone healing. However, the physiological mechanism by which they produce their therapeutic effect is not fully understood. Furthermore, there are numerous application protocols in the literature that make it very difficult to know the number of sessions, their interval or the intensity at which shock waves should be applied. This is why more work is needed to present clear methodologies and results and to help standardize the guidelines for the application of the technique.

### 3.2. Method of Application

Shock waves should be applied to the affected region. Palpatory techniques or ultrasound localization can be used to localize the segment to be treated. Both options appear to be equally beneficial [24].

Application protocols may vary according to the studies reviewed [25]. Table 2 provides guidance on recommended the intensities, pulses and number of sessions for the main indications. The intensity of the shock waves at the focal point is measured in energy flux density (EFD; mJ/mm^2^) per pulse and plays a role in the therapeutic effect of the technique [26]. In routine clinical practice, energy flux density levels vary from 0.001 to 0.5 mJ/mm^2^ [27]. To facilitate the regenerative effect on tissues, an EFD < 0.2 mJ/mm^2^ is recommended [28]. Most protocols use three shock wave sessions with a one-week break in between. The number of pulses per session usually ranges from 800 to 3000 [9]. In general, tendinopathies and osteoarthritis can be treated every two weeks, with 2000 pulses per session and four sessions in total; trapezius myofascial syndrome and low back pain with six sessions and two sessions per week and 1000 pulses per session; and delayed bone healing and avascular necrosis of the hip with two sessions in total, one every two weeks, and 4000 pulses.

It seems that in order to achieve their analgesic effect, shock wave treatment protocols should last longer than one month. In addition, it appears that shock waves, depending on the pathology, may have a greater analgesic effect when applied as monotherapy versus when applied in conjunction with other interventions [29]. 

Shock waves induce a tissue repair response, so they require time to achieve their clinical effect [30]. Clinical improvement from shock wave therapy has been reported to occur within 3–12 weeks after treatment, with benefits persisting for up to two years compared to a placebo [31]. Haake et al. found that shock waves applied under local anesthesia had no more effect than a placebo at 12 weeks [32]. Other studies, such as that of Furia et al., show similar results [33]. Therefore, although it is an uncomfortable procedure, it is recommended that shock waves are applied without local anesthesia.

It is recommended that the effectiveness of the technique be assessed at least four months after treatment, as one of its advantages is that its therapeutic effect is prolonged in the long term, more so than other types of therapy [34]. For example, the work of Ozturan et al. studied patients with epicondylitis of more than six months’ duration who were infiltrated with corticosteroids or autologous blood or received shock wave therapy. At four weeks, the corticosteroid-infiltration had the greatest effect, while at 52 weeks, the greatest effect was achieved with shock waves (89%), followed by autologous blood (83%) and, finally, corticosteroid injection (50%) [35].

It appears that the best results with shock wave therapy are achieved in patients younger than 60 years and with a symptom duration of less than 12 months [36]. When assessing the clinical involvement and treatment outcomes of MSK injuries, the two most important aspects are pain and function, as these are the most important elements for patients and are considered to be disease-specific [37].

In addition to clinical monitoring, ultrasound can be used to assess changes in tendon morphology and thickness, calcification or neovascularization. However, there may be a discrepancy between clinical and ultrasonographic findings [38]. The effectiveness of shock wave therapy for the following pathologies will be discussed below: tendinopathies, plantar fasciitis, axial pain, osteoarthritis, bone disease and entrapment neuropathies. In addition, Table 3 classifies the different conditions presented by the strength of evidence supporting the effectiveness of shock wave therapy.

### 3.3. Role of ESWT in Tendinopathies

Complete recovery from chronic tendinopathy is estimated to be around 80% [17]. Ischemia is an important etiological factor in tendinopathies, as tendons are hypovascular at the proximal insertion site, and this poor vascularization can lead to hypoxic degenerative changes when overuse occurs [39]. In addition, tendon degeneration causes fibers to become disorganized and type I collagen to be replaced by weaker type III collagen, resulting in pain and reduced tendon strength [40]. Repeated stress on the tendon results in cumulative micro-trauma. When the repair capacity of the tendon is exceeded, the sheath can be affected, causing it to degenerate [41].

Histologically, tendinopathy is characterized by the absence of inflammatory cells, intra-tendon collagenous degeneration, the thinning and disorientation of collagen fibers, neovascularization and hypercellularity with high concentrations of glycosaminoglycans and proteoglycans [17]. Using 2D ultrasonography, it has been observed that shock waves provoke reactions in fibroblasts that repair tendon cracks [42].

The tendinopathies for which ESWTs have shown efficacy are described below.

#### 3.3.1. Calcific Tendinopathy of the Shoulder

Calcium deposition in the tendons of the shoulder is a common finding that can occur in up to 7.8% to 13.6% of asymptomatic patients and in 33.3% to 42.5% of symptomatic patients [43]. Calcifying tendinopathy of the shoulder requires imaging findings of tendon calcification and compatible clinical findings: pain near the greater shoulder tuberosity, limited mobility and nocturnal discomfort [44]. High-intensity shock wave therapy is recommended for the treatment of calcific tendinopathy of the shoulder [45,46]. In this case, shock wave treatments where the calcification is localized via radiology seem to be more effective [47]. In addition, the supine position with the shoulder to be treated in hyperextension and internal rotation is recommended because it appears to be more beneficial than the neutral position [48].

A systematic review analyzing 18 articles (more than 1600 patients) reported an improvement in pain and function with shock waves versus a placebo and other conservative treatments such as TENS (transcutaneous electric nerve stimulation) or physical exercise at three and six months. Intervention with ultrasound-guided percutaneous irrigation appeared to offer better results in the pain and radiological progression of calcification at 1 year [46]. In a clinical trial of 42 patients with calcific tendinopathy of the shoulder, shock wave therapy was compared with the conservative treatment of physiotherapy. Greater benefit in terms of pain, function, quality of life and ultrasonography was reported in the shock wave group at 6 and 12 weeks after intervention [49]. For rotator cuff disease with or without calcification, a systematic review concluded that the wide clinical diversity and different treatment protocols precluded any potential benefit. It would be desirable to establish a standard dose and treatment protocol before further research [25].

#### 3.3.2. Lateral Epicondylitis

Lateral epicondylitis is a very common musculoskeletal pathology due to repeated microtrauma to the lateral epicondylar area of the elbow, usually due to overuse [50]. In addition to pain, grip strength is a measure that indicates the severity of the disease and its functional impact. In a meta-analysis including 13 articles with a total of 1035 patients, patients who received shock wave therapy had an improvement in pain and grip strength and recovered earlier compared to those who received a placebo and other conservative treatments such as cryo-ultrasound or laser [51].

#### 3.3.3. Greater Trochanteric Pain Syndrome

Greater Trochanteric Pain Syndrome is a syndrome characterized by pain in the lateral aspect of the hip that worsens with walking and lateral decubitus on the affected side. It can be caused by tendinopathy at the level of the gluteus medius or gluteus minimus, trochanteric bursitis or external coxa saltans, and is usually self-limiting or improves with conservative treatment in 90% of cases [52].

Shock wave therapy is reserved for cases where conservative treatment has not improved the clinical picture. A meta-analysis including 13 randomized clinical trials (1034 patients) found that both platelet-rich plasma and shock wave therapy produced a short-term pain benefit (one to three months). Physical therapy with a structured exercise program produced a short-term (one to three months) functional improvement [53]. Heaver et al. also reported the benefits of shock wave therapy in a randomized clinical trial in which 104 patients were treated in two groups: one group receiving extracorporeal shock wave therapy and the other group receiving ultrasound-guided corticosteroid injection therapy. At 12 months, the group receiving shock wave treatment had a greater improvement in pain (visual analog scale (VAS) 37.1 versus 55.0) and function [54].

#### 3.3.4. Patellar Tendinopathy

Knee soft tissue injuries are often due to tendinopathy or ligament injuries, causing pain and functional limitation that affect gait, running and even quality of life [55]. Knee injuries account for 35% of overuse-related sports injuries [56].

A review involving 19 randomized clinical trials found moderate evidence that shock waves significantly reduced pain by an average of 1.49 points on the visual analogue scale compared to different control groups. This effect applied both when shock waves were applied at high energy and at low energy [28]. In a recently published systematic review and meta-analysis, it was seen that shock wave treatment has little short-term effect compared to a placebo or a placebo and eccentric exercises. However, it does seem that shock waves have a significant effect on pain compared to conservative treatment [57].

Shock waves also seem to have an effect on parameters related to sports performance in the case of patellar tendinopathy. Shock wave treatment could increase vertical jump distance compared to placebo treatment [58].

#### 3.3.5. Achilles Tendinopathy

Achilles tendinopathy is characterized by a clinical picture of pain, inflammation and functional limitation, and occurs more frequently in men, probably due to their higher levels of physical activity [59]. It is classified as insertional (usually affecting the region of the posterosuperior protuberance of the calcaneus) or non-insertional (referring to symptoms 2–6 cm proximal to the insertion on the calcaneus) [60].

Numerous conservative treatments are used for Achilles tendinopathies such as ultrasound, electrotherapy, laser, orthoses and exercises. These treatments can be combined with shock waves with good results. For example, it seems that the combination of shock waves and eccentric exercises produces better results than eccentric exercises alone [61]. Also, dietary supplementation (arginine-L-alpha-ketoglutarate and hydrolyzed collagen type I) in combination with shock wave treatment produces better results than shock waves alone [62]. A recent systematic review and meta-analysis showed that shock waves may have a small effect on pain and function in the short term compared with conservative treatments such as eccentric exercises, low-laser therapy or corticosteroid injection. Furthermore, compared with shock waves, a placebo could improve some results in function but not pain [57].

### 3.4. Role of ESWT in Plantar Fasciitis

Plantar fasciitis is a clinical diagnosis that involves pain in the heel, which may affect the inner or outer area of the heel and increases with standing and walking [63]. Its pathophysiological mechanism is usually repeated micro-tears in the plantar fascia that exceed the reparative capacity of the tissues, and it usually manifests at the level of the calcaneal fat pad, calcaneal bursa, plantar aponeurosis or calcaneal hypertension [64]. It most commonly affects men in a 2:1 ratio, and healing time usually ranges from 6 to 18 months [65]. A meta-analysis analyzing 13 articles (1185 patients) found that patients treated with shock wave therapy had greater pain reduction and functional improvement and shorter return-to-work times than different control groups [66].

### 3.5. Role of ESWT in Axial Pain

Shock waves have been shown to be useful in the following problems that cause axial pain.

#### 3.5.1. Myofascial Pain Syndrome of the Trapezius

Myofascial pain syndrome is defined as a picture of regional pain with the presence of painful trigger points or taut bands with selective pain on palpation [67]. Some works try to explain the biological effect that shock waves have on myofascial syndrome. For instance, in a study in which tissue samples were collected from the fascia lata of three volunteers and subjected to 100 shock wave pulses at 0.05 mJ/mm^2^, the authors found that shock waves immediately produced hyaluronan-rich vesicles, collagen-I and collagen-III, increasing after four hours and maintaining it after 24 h, so they concluded that it could have a role in the regulation of the extracellular matrix [68]. 

A meta-analysis including 10 randomized clinical trials (477 patients) reported that shock waves had a greater effect on pain than a placebo or ultrasound, with similar effectiveness to other techniques, such as laser, dry needling or trigger point infiltration, on pain or functional improvement [69].

#### 3.5.2. Low Back Pain

In a three-month prospective randomized study (40 patients), shock wave therapy was compared with a placebo (both patients performed a 45 min exercise program five days a week). It was found that the use of shock waves together with an exercise program appeared to be effective in improving pain in patients with chronic low back pain, although it did not achieve functional improvement [70].

#### 3.5.3. Coccydynia

Coccydynia is characterized by pain in the coccyx and/or coccygeal joints that worsens with prolonged sitting and occurs most frequently after trauma [71]. In a case series of 34 patients (29 women, 5 men) with coccygodynia who underwent shock wave therapy, a statistically significant improvement in pain (from VAS 9.6 ± 0.5 on average to VAS 3 ± 3.2) was found at 6 months [72]. Another study in which four shock wave sessions were performed in 10 patients with chronic coccygodynia found an improvement in pain at four weeks and two months, but no benefit was found seven months after the last session compared to the baseline [73].

### 3.6. Role of ESWT in Knee Osteoarthritis

Degenerative osteoarthritis is the most common MSK disease. Its pathophysiology involves a series of cellular reactions involving inflammatory mediators, cytokines and matrix degradation. This results in damage to cartilage, the synovial membrane, subchondral bone, ligaments and joint muscles, leading to pain, joint limitation and functional restriction [74].

A meta-analysis involving 32 randomized clinical trials (2408 patients) found that shock waves achieved a greater reduction in pain and function than a placebo, oral medication, ultrasound, hyaluronic acid (although with heterogeneous studies), intra-articular corticosteroids and platelet-rich plasma (although with no difference in function) [75]. In a randomized clinical trial of 125 patients with knee osteoarthritis undergoing shock wave therapy, it was reported that moderate intensity EFD (0.12 to 0.25 mJ/mm^2^) seems to have better results in terms of pain and function, with no difference found between receiving 2000 and 4000 pulses per session [76].

### 3.7. Role of ESWT in Bone Diseases

Shock waves have been shown to be useful in the following bone conditions.

#### 3.7.1. Fracture Nonunion

The effects of shock waves at the level of consolidation became known when in 1988, it was shown that patients treated with shock waves during lithotripsy had an increased pelvic osteogenic response. It is now used successfully, and radiological guidance is recommended to localize the application site [77]. In a randomized clinical trial with 126 long-bone nonunion patients, three groups were assigned: group one received extracorporeal shock wave treatment with an EFD of 0.40 mJ/mm^2^; group two received extracorporeal shock wave treatment with an EFD of 0.70 mJ/mm^2^; and group three received surgical treatment. The clinical and radiographic outcomes were assessed for up to 24 months, and all three treatment groups were found to have similar results [78]. The positive results of shock waves in the treatment of consolidation delays are also evidenced by a meta-analysis of 1737 patients with symptoms including nonunions and consolidation delays of both the long bones and small bones of the hands and feet, in which success rates of 62% to 83% are reported with ESWT [79].

It appears that the effectiveness of shock waves in the treatment of bone consolidation problems depends on the type of nonunion. Hypertrophic long bone nonunions have an improvement rate of 80% to 100% with shock waves, while atrophic nonunions have a lower response rate estimated to be around 23% to 27% [80].

#### 3.7.2. Femoral Head Osteonecrosis

Femoral head osteonecrosis is a disease with an unknown pathogenesis that can occur in different body regions and is characterized by a pathological increase in interstitial fluid, probably secondary to a vascular reaction to an internal or external process [81]. In some cases, such as osteonecrosis of the femoral head, if left untreated, it can progress to joint collapse requiring total hip arthroplasty [82]. In this regard, it appears that shock wave therapy in the early stages may help to prevent the progression of the area of avascular necrosis [83].

A meta-analysis including six randomized clinical trials (256 patients) reported that shock wave therapy appears to produce an improvement in pain and function in osteonecrosis of the femoral head, especially in the early stages (ARCO stage I and ARCO stage II), and the clinical benefits are maintained in the long term. Adding treatment with prostaglandin inhibitors or bisphosphonates does not seem to improve outcomes [84]. However, Sconza et al., in a systematic review including five studies (199 patients), did observe that the combined use of bisphosphonates with extracorporeal shock waves in the treatment of osteonecrosis had a synergistic effect, although they reported low validity of their results due to the low quality of the studies analyzed [85].

#### 3.7.3. Kienbock’s Disease

In a case series with 22 patients, shock wave therapy was found to offer an improvement in pain, mobility and magnetic resonance imaging (MRI) after 60 days [86].

#### 3.7.4. Pubis Osteitis

In a study involving 44 athletes (all of whom received intensive rehabilitation treatment), shock wave therapy was found to improve pain and function at one and three months compared to a placebo. In addition, the treatment group was able to resume sporting activity (football) earlier (73.2 days versus 102.6 days) [87].

#### 3.7.5. Bone Marrow Edema Syndrome of the Hip

In a retrospective study of 46 patients who had received conservative treatment, surgical core decompression or shock wave treatment was assigned. All patients recovered clinically at 12 weeks with no pathological findings on MRI at six months, but the shock wave treatment group had a greater improvement in pain and resumed daily activities earlier [88]. Another retrospective study with 20 patients also reported functional and MRI improvements at two months that increased up to six months [81].

### 3.8. Role of ESWT in Carpal Tunnel Syndrome

Surgical intervention of the median nerve in the carpal tunnel by releasing the flexor retinaculum is the definitive treatment for carpal tunnel syndrome and is effective in the medium and long term [89].

A systematic review analyzing 10 studies involving 433 patients (501 wrists) found that shock waves would be effective in improving pain, other symptoms and function in patients with carpal tunnel syndrome. Radial shock waves would have better results than focal shock waves [90]. A meta-analysis including five randomized clinical trials (204 patients) compared shock wave therapy and local corticosteroid infiltration therapy. Both interventions were reported to have similar effects in terms of improving pain, function and electrophysiological nerve parameters [91]. Another meta-analysis of seven clinical trials (376 patients) reported that adding shock wave therapy to a conservative treatment intervention such as a nocturnal wrist splint produced a transient improvement in pain and functional benefit for only four weeks [92].

### 3.9. Strengths and Weaknesses of the Studies 

Although the evidence supporting the effectiveness of shock waves seems clearly favorable, there are some limitations in the studies that should be improved. As mentioned previously, there is a lack of unified protocols to apply to the same clinical indications and to be able to draw solid conclusions. Furthermore, many studies compare shock waves with different conservative treatments. Due to the variability in the conservative treatments used in different studies (laser, ultrasound, TENS, different exercise programs, etc.) it is difficult to establish the role of shock waves compared to other therapies [45,48,50,64,69,75]. Furthermore, there are few studies that have included the combined effect of conservative treatments with shock waves [61], so their role cannot be adequately evaluated. Regarding interventional treatment, there is also variability. Shock waves have been compared with infiltration with corticosteroids, hyaluronic acid or platelet-rich plasma, without clear studies comparing all these techniques in different pathologies [52,53,75]. It would also be interesting to have more studies comparing shock waves with surgical treatment, as in many cases, it is the most common clinical option when shock wave treatment fails [79,84,90].

In general, shock wave therapy has been widely studied and there are numerous high-evidence studies such as systematic reviews and meta-analyses. However, there is great heterogeneity in the application protocols and great variability in the conservative and interventional treatment techniques with which the effectiveness of the technique is compared. Furthermore, in many cases, the exact parameters of the intervention are not detailed (dose, method of application, model of shock wave generator, etc.), nor are the therapies received by the control group (ultrasound doses or repetitions, resistances and types of the exercises performed). Furthermore, given that the biological mechanisms by which shock waves produce their therapeutic effect are not yet known in depth, the quality of the evidence generated for this technique must continue to be improved.

### 3.10. Adverse Effects and Contraindications

Among the usual contraindications of shock wave treatment are anticoagulant treatment or disorders that favors bleeding (since a high-intensity shock wave can produce bleeding). Other contraindications include acute infection, pregnancy or direct application to the growth plate. Nerve tracts and large vessels should be avoided during application [93]. The vast majority of studies have found no significant adverse effects during or after shock wave application [42,45,50,52,61,66,69,70,73,75,78,84,92].

It is convenient to apply shock waves over a number of sessions and at a suitable intensity, and performing the technique with an excessive dose can increase the risk of adverse effects occurring [94] and that the treatment will be ineffective [95]. Shock wave treatments that are applied with high intensity have an increased risk of bruising [96]. Cancer itself is not a contraindication to perform shock wave treatment, although it is contraindicated in the case of metastasis, malignant tumors, multiple myeloma and lymphoma and when the area being dealt with contains oncological tissue [2].

The contraindication for tendon ruptures is not well established. In general, shock wave therapy is not recommended when there is a complete tendon rupture, but the treatment of extensive tendon injuries (>6 mm or >50%) has been performed without significant adverse effects [97]. One study reported a rupture of the Achilles tendon two weeks after a first shock wave session, although the authors did not consider this to be a consequence of this treatment. Nevertheless, imaging tests are recommended, especially in patients over 60 years of age, before shock wave treatment [98].

A summary of contraindications for shock wave therapy is shown in Table 4.

## 4. Limitations of the Study

This is a narrative review in which the articles selected were chosen as the most important in relation to the title of this article, prioritizing the studies with the most scientific evidence (meta-analyses, systematic reviews and randomized studies). 

Although there are many mechanisms of action described for shock waves, the complete mechanism is not yet fully understood, nor are the most effective application protocols. In fact, there is great heterogeneity in application protocols, which makes comparison between studies difficult. 

There are still aspects that have not been sufficiently investigated in the literature and that could help to better understand the role of shock waves in the management of musculoskeletal diseases. Although the effect of shock waves seems clear, the physiological mechanisms by which it acts are not yet fully understood. Furthermore, at a clinical level, it is necessary to have more articles that unify the application protocols to be able to carry out homogeneous studies. On the other hand, it is advisable to better understand the role of shock waves when combined with other conservative therapies. There is still a lack of high-quality articles on certain less frequent indications, and it is important to better study the safety of shock waves when there are tendon ruptures.

Further evidence on the efficacy of shock wave therapy in the treatment of musculoskeletal pathologies that cause pain is desirable in the future. It is also essential to have more data on the best treatment protocols to help apply the technique optimally. Due to the numerous mechanisms of action of shock waves, it is expected that new clinical indications in musculoskeletal pathologies will continue to emerge.

## 5. Conclusions

Extracorporeal shock waves are a non-invasive, safe and effective treatment that can be applied to a large number of MSK pathologies in which the usual conservative treatment has failed. Of the clinical indications, calcific tendinopathy of the shoulder, lateral epicondylitis, greater trochanteric pain syndrome, plantar fasciitis and delayed bone healing present a level of evidence of 1+. Patellar tendinopathy, Achilles tendinopathy, trapezius myofascial syndrome, low back pain, avascular necrosis of the hip, osteoarthritis, femoral head osteonecrosis, pubis osteitis and carpal tunnel syndrome present a level of evidence of 1−. Although their mechanism of action is not completely known, their biological effects at analgesic, osteogenic and tissue-reparative levels achieve an improvement in pain and function that can be maintained in the long term. There is still no consensus on the application protocols for the different MSK conditions, so there is some variability in the recommended number of pulses, intensity, number of sessions and frequency. It is desirable to have more articles studying the role of shock waves in MSK medicine, and for them to be carried out with a more homogeneous methodology in order to obtain more solid clinical conclusions.

## Figures and Tables

**Figure 1 healthcare-11-02830-f001:**
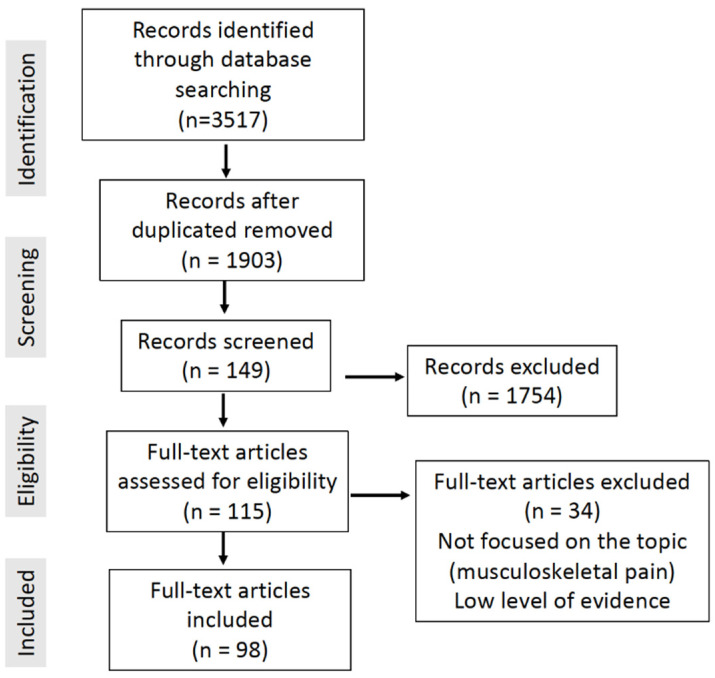
Flowchart diagram of the literature review for this article and terms “extracorporeal shock wave therapy pain”.

**Figure 2 healthcare-11-02830-f002:**
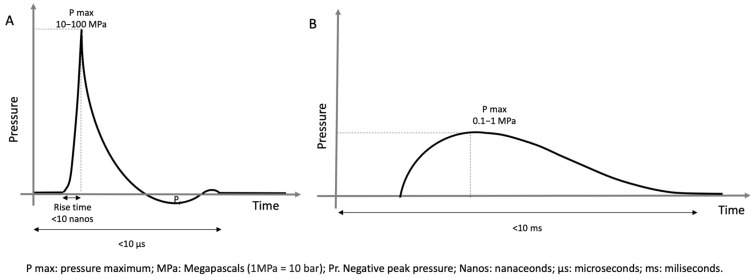
Parameters of different shock waves. (**A**) Focused shock waves. (**B**) Radial pressure waves.

**Table 1 healthcare-11-02830-t001:** Main therapeutic effects of extracorporeal shock wave therapy.

Therapeutic Effects	Biological Effects
Analgesic effect	Decreased substance P in the area of application [9]
	Selective loss of unmyelinated nerve fibers [10]
	Decreased expression of calcitonin-related peptide in dorsal root ganglia [11]
	Activation of the serotonergic system [12]
Tissue repair effect	Proliferation of tenocytes [13]
	Activation of catabolic processes leading to the elimination of damaged matrix constituents [14]
	Microdisruption of avascular or poorly vascularized tissues [15]
	Increased tissue neovascularization [16]
	Enhanced collagen synthesis, maturation and characteristics [17]
	Regulation in proliferation, activation and differentiation of keratinocytes originating from scar tissue (antifibrosis) [18]
Osteogenic effect	Osteoblast growth through osteogenic transcription factors such as vascular endothelial growth factor-A (VEGF-A) and hypoxia-inducible factor-1α [19]
	Regulation and stimulation of chondrogenesis and bone regeneration through mesenchymal stem cell metabolism [20]
	Enhancement of Pdia-3 expression involved in the 1α,25-Dihydroxyvitamin D 3 Rapid Membrane Signaling Pathway, related to calcium homeostasis [21]
	Stimulation of the periosteum with decreased osteoclast activity [22]
	Osteoblast proliferation and differentiation through regulation of nitric oxide (NO), protein kinase B (PKB), bone morphogenetic protein-2 (BMP-2) and transforming growth factor-beta 1 (TGF-β1) levels [23]

**Table 2 healthcare-11-02830-t002:** Summary of the main indications of shock waves and their recommended applications.

Pathology	Intensity	Sessions	Pulses	Comments
Calcific tendinopathy of the shoulder	High	3–4 (every 1–2 weeks)	1500–2000	Locate calcification. Patient in supine position with shoulder in extension and internal rotation
Lateral epicondylitis	Low	3 (every 1–2 weeks)	1500–2000	Apply to point of maximum pain
Greater trochanteric pain syndrome	Low	3 (every 1–2 weeks)	2000	Apply to point of maximum pain
Patellar tendinopathy	Low	3 (every 1–2 weeks)	1500–2000	Apply to point of maximum pain
Achilles tendinopathy	Low	4 (every 1–2 weeks)	2000	Apply to point of maximum pain
Plantar fasciitis	Low	3 (every 1–2 weeks)	2000	Apply to point of maximum pain
Trapezius myofascial syndrome	Low	4–8 (1–2 per week)	1000	Apply to point of maximum pain
Low back pain	Low	6–10 (1–2 per week)	1000	Apply to point of maximum pain
Delayed bone healing	High	1–4 (every 1–2 weeks)	2000–4000	Localize the area using radiology
Avascular necrosis of the hip	High	1–2 (every 1–2 weeks)	4000–6000	Locate the area using radiology
Osteoarthritis	Low	4 (every 1–2 weeks)	2000	Apply to point of maximum pain
Carpal tunnel syndrome	Low	3 (every 1–2 weeks)	1000–1500	Apply to point of maximum pain

**Table 3 healthcare-11-02830-t003:** Effectiveness of shock wave therapy for the following pathologies presented by level of evidence.

Pathologies	Level of Evidence
Calcific tendinopathy of the shoulder	1+
Lateral epicondylitis	1+
Greater trochanteric pain syndrome	1+
Plantar fasciitis	1+
Delayed bone healing	1+
Patellar tendinopathy	1−
Achilles tendinopathy	1−
Trapezius myofascial syndrome	1−
Low back pain	1−
Avascular necrosis of the hip	1−
Osteoarthritis	1−
Femoral head osteonecrosis	1−
Pubis osteitis	1−
Carpal tunnel syndrome	1−
Bone marrow edema syndrome of the hip	2−
Coccigodinia	3
Kienbock’s disease	3

**Table 4 healthcare-11-02830-t004:** Contraindications to extracorporeal shock wave therapy (ESWT) [93,94,95,96,97].

Patients with poorly controlled coagulopathies.
Acute infection
Pregnancy
Direct application on growth plate
Oncological tissue in the area to be treated
Tumor metastases
Multiple myeloma
Lymphoma
Complete tendon rupture

## Data Availability

Not applicable.

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
