# Peer review of "Extracorporeal Shock Wave Therapy for the Treatment of Musculoskeletal Pain: A Narrative Review"

_healthcare, 2023, doi:10.3390/healthcare11212830_

Round 1

Reviewer 1 Report

Reviewer's Report

Title: Extracorporeal shockwave therapy for the treatment of musculoskeletal pain.

Abstract:

1. The abstract provides a brief overview of extracorporeal shockwave therapy (ESWT) but lacks specific details about the scope and purpose of the review. It should mention the research question or objectives of the review to give readers a clear sense of what to expect.

2. The abstract states, "There is solid evidence that supports their use to improve pain in many MSK pathologies," but it does not provide any quantitative or qualitative summary of this evidence. Including key findings or a summary of the types of pathologies and their level of evidence would enhance the abstract's informativeness.

Introduction:

1. The introduction briefly introduces shockwave therapy but does not provide a clear narrative that leads to the research question or objective of the review. It should establish the need for the review and why updating the role of ESWT in musculoskeletal pain treatment is important.

2. The statement, "The aim of this article is to update the role of ESWT in the treatment of various pathologies that cause musculoskeletal (MSK) pain," is rather vague. It would be helpful to specify the key research questions or objectives that the review aims to address.

Methods:

1. The methods section lacks detail about the search strategy and inclusion criteria used in the literature search. While it mentions the databases searched and keywords used, it does not provide the actual search terms or the specific criteria for article selection. This information is crucial for the transparency and reproducibility of the review.

2. The number of identified articles in each database is mentioned but not explained. It would be beneficial to provide a brief rationale for the inclusion/exclusion of articles based on the search results.

3. The section mentions prioritizing meta-analyses, systematic reviews, and randomized studies when available, but it does not clarify how other types of studies were handled or why these study types were prioritized. Providing a clear hierarchy of evidence and explaining the rationale for inclusion criteria would enhance the methodology's transparency.

RESULTS:

1. The results section provides an extensive summary of various musculoskeletal conditions and their response to shockwave therapy. While this information is informative, it lacks a clear structure and organization. It would be helpful to present the results in a more systematic way, perhaps categorizing them by the type of condition or level of evidence to enhance readability.

2. The section discusses the mechanism of action, method of application, and therapeutic effects of shockwaves, which is valuable information. However, it lacks critical analysis and synthesis of the results presented. Readers would benefit from a concise summary of the key findings and their implications.

3. The section includes numerous references to studies without providing a clear indication of the strength of evidence or the quality of the studies. It would be helpful to provide some context or critical appraisal of the studies cited to assess the reliability of the evidence presented.

4. While the section discusses the effectiveness of shockwave therapy for various conditions, it does not adequately address any limitations or potential biases in the studies reviewed. A brief discussion of study limitations, such as sample size, study design, and potential conflicts of interest, would provide a more balanced view of the evidence.

5. The section could benefit from a concise summary of the adverse effects and contraindications of shockwave therapy, as mentioned in the text, to ensure that readers have a comprehensive understanding of the treatment's safety profile.

DISCUSSION:

1. The discussion section is quite brief and lacks an in-depth analysis of the results presented in the previous section. It should provide a critical synthesis of the findings, discussing the clinical implications and limitations of the evidence.

2. The section does not offer any insights into future research directions or areas where additional studies are needed. Discussing the gaps in the current literature and suggesting potential avenues for future research would enhance the manuscript's value.

3. There is a need for more critical appraisal of the quality of the studies reviewed. It would be beneficial to discuss the strengths and weaknesses of the available evidence, such as the risk of bias in individual studies and the overall quality of the evidence.

4. The discussion could be structured more coherently to follow the same organization as the results section. This would make it easier for readers to connect the findings with their interpretation and implications.

5. While the section mentions the variability in application protocols, it could delve deeper into the implications of this variability for clinical practice. How should clinicians decide on the appropriate protocol for a specific patient or condition? Addressing these practical considerations would provide more guidance to healthcare professionals.

nil

Author Response

Dear reviewer, 

Thank you very much for your review and for all your contributions to this article. We will comment below on all the changes made according to your suggestions and how they have been introduced in the text. The changes made according to your suggestions can be found in the attached file and have been added to the text in blue.

Warm regards

Reviewer 2 Report

This manuscript describes aims to provide an update on the use of shockwave therapy in MSK conditions. It is well written. A few comments below:

1. The introduction requires some elaboration on the current literature, in light of the results of your study. You can add a paragraph about the possible mechanisms of action and the various current indications.

2. In your conclusions, please refer to the strength of evidence found for each class of indications.

Author Response

Dear reviewer, 

Thank you very much for your review and for all your contributions to this article. We will comment below on all the changes made according to your suggestions and how they have been introduced in the text. The changes made according to your suggestions can be found in the attached file and have been added to the text in red.

Warm regards

Reviewer 3 Report

Title

Title is not appropriate because it is not completely informative about the contents of the paper, should be better specified the type of review. Modify the title.

Abstract

The abstract respects the rules of the journal. The background and the aim are interesting. In the design is not present the type of study: what type of review? Methods need to be better explained. The clinical Impact of this review is present but need to be better explained.

Text

The introduction of the study does not clearly sum up the background of the study, but it needs to be improved by adding some reference. The authors provide a rationale for performing the study based on a review of the medical literature, but they need to better define it. Furthermore, they define well terms used in the remainder of the manuscript. The hypothesis needs to better be defined.

The methods are not clear. What type of review is? Only at the end it is mentioned that it is a narrative review.

In the flow chart in the methods needs explain why articles were excluded.

Who performed the review? Explain how the different co-authors participated to review. The methods are not clear about the procedures of review. Add a more precisely description of them.

The study needs to be better explained in the methodology.

The number of references reported about the topic must be increased:

-         speaking of myofascial pain syndrome, references should be added about the possible mechanism to solve the syndrome. The authors provide a rationale for this evaluation on a review of the medical literature, but the number of references reported about this topic must be increased. For example, immediate effects of fESWs on deep fasciae can be mentioned, by citing the following papers:” Immediate Effects of Extracorporeal Shock Wave Therapy in Fascial Fibroblasts: An In Vitro Study. Pirri C, Fede C, Petrelli L, De Rose E, Biz C, Guidolin D, De Caro R, Stecco C. Biomedicines. 2022 Jul 18;10(7):1732..”. 

-         the same thing speaking of the patellar and Achilles tendinopathy: The effectiveness of shockwave therapy on patellar tendinopathy, Achilles tendinopathy, and plantar fasciitis: a systematic review and meta-analysis.  Charles R, Fang L, Zhu R, Wang J. Front Immunol. 2023 Aug 16;14:1193835.”.

The results are reported clearly and concisely.

Add a discussion paragraph.

References

They are qualified and updated with the lasted data, but they must be integrated. The reference list follows the format for the journal.

Tables

They highlight the key points.

Figures

They highlight the key points but they need to be improved. Delete Figure 3, it is not functional to the paper.

Statistical Analysis

It isn’t needed further checking of data by a statistician reviewer.

General comments

The purpose of the study is original but the study needs to be improved in the introduction, methods and discussion. The hypothesis needs to better be defined. The methods are not clear. The study has been structured and carried out correctly but the methodology needs to be better explained. Discussion needs to be added.

Title

Title is not appropriate because it is not completely informative about the contents of the paper, should be better specified the type of review. Modify the title.

Abstract

The abstract respects the rules of the journal. The background and the aim are interesting. In the design is not present the type of study: what type of review? Methods need to be better explained. The clinical Impact of this review is present but need to be better explained.

Text

The introduction of the study does not clearly sum up the background of the study, but it needs to be improved by adding some reference. The authors provide a rationale for performing the study based on a review of the medical literature, but they need to better define it. Furthermore, they define well terms used in the remainder of the manuscript. The hypothesis needs to better be defined.

The methods are not clear. What type of review is? Only at the end it is mentioned that it is a narrative review.

In the flow chart in the methods needs explain why articles were excluded.

Who performed the review? Explain how the different co-authors participated to review. The methods are not clear about the procedures of review. Add a more precisely description of them.

The study needs to be better explained in the methodology.

The number of references reported about the topic must be increased:

-         speaking of myofascial pain syndrome, references should be added about the possible mechanism to solve the syndrome. The authors provide a rationale for this evaluation on a review of the medical literature, but the number of references reported about this topic must be increased. For example, immediate effects of fESWs on deep fasciae can be mentioned, by citing the following papers:” Immediate Effects of Extracorporeal Shock Wave Therapy in Fascial Fibroblasts: An In Vitro Study. Pirri C, Fede C, Petrelli L, De Rose E, Biz C, Guidolin D, De Caro R, Stecco C. Biomedicines. 2022 Jul 18;10(7):1732..”. 

-         the same thing speaking of the patellar and Achilles tendinopathy: The effectiveness of shockwave therapy on patellar tendinopathy, Achilles tendinopathy, and plantar fasciitis: a systematic review and meta-analysis.  Charles R, Fang L, Zhu R, Wang J. Front Immunol. 2023 Aug 16;14:1193835.”.

The results are reported clearly and concisely.

Add a discussion paragraph.

References

They are qualified and updated with the lasted data, but they must be integrated. The reference list follows the format for the journal.

Tables

They highlight the key points.

Figures

They highlight the key points but they need to be improved. Delete Figure 3, it is not functional to the paper.

Statistical Analysis

It isn’t needed further checking of data by a statistician reviewer.

General comments

The purpose of the study is original but the study needs to be improved in the introduction, methods and discussion. The hypothesis needs to better be defined. The methods are not clear. The study has been structured and carried out correctly but the methodology needs to be better explained. Discussion needs to be added.

Author Response

Dear reviewer, 

Thank you very much for your review and for all your contributions to this article. We will comment below on all the changes made according to your suggestions and how they have been introduced in the text. The changes made according to your suggestions can be found in the attached file and have been added to the text in green.

Warm regards

Reviewer 4 Report

This study is a review paper on the effect of the treatment of musculoskeletal pain of extracorporeal shock wave therapy. 

The contents of the review are as follows.

- I think it would be good if you could reveal a native review in the title.

- Please clarify the distinction between sentences and paragraphs. Sentences are often paragraphed throughout the paper. All of those parts must be modified.

This study describes the overall status of ESWT and is worth publishing.

Author Response

Dear reviewer, 

Thank you very much for your review and for all your contributions to this article. We will comment below on all the changes made according to your suggestions and how they have been introduced in the text. The changes made according to your suggestions can be found in the attached file.

Warm regards

Round 2

Reviewer 1 Report

I believe the authors have effectively responded to the my comments in the revised manuscript, so I have no additional remarks to add. The manuscript can be accepted in tis current form.